# Procalcitonin Values Fail to Track the Presence of Secondary Bacterial Infections in COVID-19 ICU Patients

**DOI:** 10.3390/antibiotics12040709

**Published:** 2023-04-05

**Authors:** Elsa Harte, Sharuha Kumarasamysarma, Benjamin Phillips, Olivia Mackay, Zohra Rashid, Natalya Malikova, Abdullah Mukit, Saraswathi Ramachandran, Anna Biju, Kate Brown, Rosie Watts, Charlie Hodges, William Tuckwell, Nick Wetherall, Henry Breen, Shannon Price, Tamas Szakmany

**Affiliations:** 1School of Medicine, Cardiff University, Cardiff CF14 4XN, UK; 2Department of Anaesthesia, Intensive Care and Pain Medicine, Cardiff University, Cardiff CF14 4XN, UK; 3Critical Care Directorate, Grange University Hospital, Aneurin Bevan University Health Board, Cwmbran NP44 2XJ, UK

**Keywords:** COVID-19, procalcitonin, secondary infection, antimicrobial stewardship, antibiotics, immunomodulators

## Abstract

The development of secondary bacterial infections in COVID-19 patients has been associated with increased mortality and worse clinical outcomes. Consequently, many patients have received empirical antibiotic therapies with the potential to further exacerbate an ongoing antimicrobial resistance crisis. The pandemic has seen a rise in the use of procalcitonin testing to guide antimicrobial prescribing, although its value remains elusive. This single-centre retrospective study sought to analyse the efficacy of procalcitonin in identifying secondary infections in COVID-19 patients and evaluate the proportion of patients prescribed antibiotics to those with confirmed secondary infection. Inclusion criteria comprised patients admitted to the Grange University Hospital intensive care unit with SARS-CoV-2 infection throughout the second and third waves of the pandemic. Data collected included daily inflammatory biomarkers, antimicrobial prescriptions, and microbiologically proven secondary infections. There was no statistically significant difference between PCT, WBC, or CRP values in those with an infection versus those without. A total of 57.02% of patients had a confirmed secondary infection, with 80.2% prescribed antibiotics in Wave 2, compared to 44.07% with confirmed infection and 52.1% prescribed antibiotics in Wave 3. In conclusion, procalcitonin values failed to indicate the emergence of critical care-acquired infection in COVID-19 patients.

## 1. Introduction

The Director-General of the World Health Organisation announced that the Severe Acute Respiratory Syndrome Coronavirus 2 was a global pandemic on 11 March, 2020 [1]. In the subsequent 35 months, it has contributed to 757,264,511 cases of COVID-19, including 6,850,594 deaths [2]. The virus has demonstrated heterogeneity in its clinical presentation. While the most common symptoms include fever, myalgia and cough [3], a subset of patients’ cases advance to multi-organ failure, shock or death [4,5,6]. 

The development of secondary bacterial infection in COVID-19 patients has been linked to worse clinical outcomes and increased mortality [7]. Early clinical guidance encouraged the administration of empirical antibiotics in hospitalised COVID-19 patients, irrespective of a confirmed bacterial co-infection [8]. Studies have subsequently revealed a significant disparity between the high rate of antimicrobial therapy and the low prevalence of confirmed secondary bacterial infections [9,10].

The difficulty in recognising a true co-infection of bacterial aetiology may have resulted in an unnecessary increase in antimicrobial prescribing [11,12], further exacerbating an antimicrobial resistance crisis that was considered a threat to modern medicine even prior to the pandemic [13]. The clinical community proposed further research into the various methods of identifying secondary bacterial infections in COVID-19 patients, thus determining optimal strategies of antimicrobial prescribing.

Procalcitonin (PCT) has previously been used to differentiate viral from bacterial infections [14]. However, its ability to predict the presence of underlying bacterial infections and guide microbiological therapy in COVID-19 patients is still uncertain [7].

Despite this uncertainty, the first wave of the pandemic saw an increase in PCT use from 48% to 84.4% in UK Intensive Care Units [15]. The self-reported use of PCT testing was to mitigate the over-administration of empirical antibiotics [15]. The efficacy of its implementation is conflicting [16]. Despite a general reduction in antibiotic administration with the use of PCT-guided prescribing [16,17,18,19], there is a remaining uncertainty as to whether PCT values can reveal the presence of bacterial infections in COVID-19 patients [7,20].

Although most clinical trials used absolute PCT values to try to diagnose infection and limit antibiotic exposure, the change in PCT value (deltaPCT) has been shown to be a more sensitive marker of infection [21]. In a small cohort of patients admitted during the first wave of the pandemic, we found that deltaPCT might be useful in detecting secondary infections [22].

This study aims to analyse the efficacy of absolute and delta procalcitonin values in tracking the presence of microbiologically confirmed secondary infections in COVID-19-positive ICU patients throughout the second and third waves of the pandemic.

The study will also analyse other biomarkers of inflammation, such as white blood cell (WBC) counts and C-Reactive Protein (CRP) values.

Secondary objectives involve investigating the antimicrobial prescribing practices throughout the pandemic with a comparison of antimicrobial prevalence to microbiologically confirmed secondary infection.

## 2. Results

### 2.1. Infection Data

Based on the inclusion criteria, 238 patients were analysed: 121 patients in Wave 2 and 117 in Wave 3. Of these patients, 119 (51.1%) tested positive for an infection (5 patients in Wave 3′s infection status were unavailable). Patient demographics, comorbidities, initial Sequential Organ Failure Assessment (SOFA) Scores, length of stay in the ICU, and mortality are summarised in Table 1. All patients were admitted from their homes, except for 13 patients (5.9%) who were transferred from other ICUs as a capacity transfer. The median length of hospital stay was 3 (1–5) days before ICU admission. 

The mean age was 55.91 (SD 13.37). The hospital mortality was 40.3%. Mortality was higher (56 (47.1%) of 119) in the Infection Group than in the No Infection Group (38 (33%) of 114). 

The median Length of ICU stay was higher in the Infection Group (12.3 [IQR 7.9–21]) compared to the No Infection Group (5 [2.9–7.4]). The mean 1st day of infection was Day 6 (SD 4). All patients received dexamethasone and a total of 117 patients received IL-6R inhibitors (49 patients had tocilizumab and 68 patients had sarilumab).

The prevalence of the causative pathogens is summarised in Figure 1. 

The most common pathogen found in positive sputum culture results was *Staphylococcus aureus* (SAUR) (*n* = 39). The most common pathogens associated with a bloodstream infection were *Escherichia coli* (ECOL) (*n* = 3) and *Staphylococcus aureus* (SAUR) (*n* = 3).

### 2.2. Inflammatory Markers 

The median Procalcitonin levels (ng/mL) on the day of ICU admission were 0.235 (IQR 0.1025–0.7750) in the No infection Group, compared to 0.18 (0.1–0.54) in the Infection Group.

PCT values for the first 15 days are detailed in Table 2 and demonstrated in Figure 2. There was no significant difference compared to baseline values or between the Infection and No Infection groups at any time points. (Appendix B and Appendix C).

We examined if PCT change from the day before microbiology testing was performed in response to clinical suspicion of infection to the day when samples were taken could predict the development of infection.

In this respect, we compared the deltaPCT values between cohorts (W = 307, *p* = 0.1263) which indicated no significant difference; therefore, we have not formally tested the predictive capabilities of deltaPCT.

Looking at other commonly used infection markers, the white blood cell counts and C-reactive protein (CRP) values did not demonstrate any significant differences between cohorts (Figure 3 and Figure 4). 

Notably, Sequential Organ Failure Assessment (SOFA) scores were significantly higher in the Infection Group on Days 0–4 compared to the No Infection Group (see Appendix D and Figure 5).

### 2.3. Antimicrobial Use across Waves 2 and Waves 3

In total, 57.02% of patients from Wave 2 had a confirmed infection and 80.2% of the Wave 2 cohort were prescribed antibiotics. The antibiotics density for Wave 2 averaged 0.92. 

A total of 44.07% of the patients admitted during Wave 3 had a laboratory-confirmed infection, and 52.1% of the cohort were prescribed antibiotics. In Wave 3, there was significantly reduced usage of antibiotics, further demonstrated in Figure 6.

Figure 6 shows the frequency of varying density of antibiotic use in the Wave 2 (A) and Wave 3 (B) cohorts.

Although not vastly different from each other, a higher proportion of the cohort had fewer cumulative days on antibiotics. A total of 90 patients had 0–1 density in Wave 3 compared to 84 in Wave 2.

The median length of antibiotic therapy (LOT) was greater in Wave 2 than Wave 3 (Wave 2: 5 days [IQR 1–10]; Wave 3: 1 day [IQR 0–5]). However, LOT was lower than ICU LOS in both waves (Wave 2: 8 days [4, 17]; Wave 3: 7 days [4, 12]) (Figure 7).

Piperacillin/tazobactam (Tazocin) was the most commonly prescribed antibiotic in both cohorts (Figure 8), with higher use in Wave 2 (*n* = 27) than in Wave 3 (*n* = 18).

Piperacillin/tazobactam is recommended by NICE as the first-line antimicrobial if the patient has severe symptoms, signs of sepsis, or is at higher risk of resistance [23].

The second most common combination of initial antimicrobials was Amoxicillin and Clarithromycin (Figure 8), recommended by NICE Guidelines in suspected moderate community-acquired pneumonia [24].

## 3. Discussion

The results of our single-centre observational study of 238 patients demonstrated no significant differences in absolute or delta PCT levels, WBC count, or CRP values between those COVID-19 patients with secondary bacterial infection and those without. In our cohort, 57.02% and 44.07% of patients from Wave 2 and 3 had a confirmed infection, while 80.2% and 52.1% were prescribed antibiotics in Wave 2 and 3, respectively.

The current NICE COVID-19 Guidance NG191 emphasises the current insufficiency of evidence to recommend routine PCT testing as a method of guiding antimicrobial prescribing [25]. Our results demonstrating the similarity in PCT levels between infected and non-infected cohorts further adds to the evidence and suggest its use does not add benefit in identifying bacterial infections and subsequent decisions regarding antimicrobial prescribing.

Despite this guidance, PCT testing has increased significantly in NHS Hospitals to guide antimicrobial decisions since the onset of the pandemic [15]. This has been due to its reported accuracy and superiority over CRP in determining the diagnosis of bacterial infections in hospitalised patients [26]. Our results do not support the applicability of PCT or CRP testing to indicate the emergence of bacterial infection in the ICU population. Interestingly, we have previously observed a positive correlation between patients experiencing a PCT increase and the occurrence of an ICU-acquired infection (*p* = 0.021) during the first wave of the pandemic [22]; however, our current results could not substantiate this finding.

Many clinicians place weight on the absolute PCT levels when determining the presence or absence of bacterial infections. In a retrospective study of 99 patients, Pink et al. demonstrated a 94% negative predictive value for detecting secondary bacterial infection in COVID-19 patients with procalcitonin values of less than 0.55 ng/mL [27]. In our cohort, the median PCT levels were well below this threshold at all time points, indicating that reliance on absolute PCT values is probably misleading in this patient cohort.

Our results emulate Heer and colleagues’ work [20], wherein their retrospective observational study (*n* = 60) demonstrated that PCT concentrations did not differ between those with positive bacterial cultures and those without (*p* = 0.10).

Overall, there is a relative paucity of evidence analysing the efficacy of PCT as an identifier of potential bacterial secondary infection in the ICU, or as co-infection in COVID-19 patients, specifically. A meta-analysis of 14 studies examining the recognition of secondary bacterial infections in COVID-19 patients and its correlation with PCT concluded that while increased PCT values may be able to identify a subset of patients at increased risk of worse outcomes, further studies are needed to evaluate its use in determining secondary bacterial infections and consequently guiding antimicrobial therapy [7].

During the pandemic, PCT-guided antimicrobial prescribing resulted in reduced empirical antibiotic use [16,17,18,19] without negatively impacting patient clinical outcomes such as mortality [17,19] and length of stay [19]. An important limitation of these studies is their omittance of the subsequent proportion of these patients assigned to the high and low PCT groups that had a confirmed bacterial infection. Considering a raised PCT may be reflective of an underlying hyperinflammatory response [28,29] and a low PCT may be a consequence of prevalent concomitant treatments throughout the pandemic, such as corticosteroids and IL-6 inhibitors [30], the results of these studies fail to fully demonstrate whether the use of PCT testing was effective in determining those patients with a concurring secondary infection and, thus, requiring antimicrobial treatment.

Importantly, our results demonstrate the relative discrepancy in the proportion of COVID-19 patients receiving antibiotics compared to those with an actual confirmed bacterial infection (Wave 2: 57.02% with infection vs. 80.2% prescribed antibiotics; Wave 3: 44.07% with infection vs. 52.1% prescribed antibiotics). These inconsistencies are replicated in the results of the International Severe Acute Respiratory and Emerging Infections Consortium (ISARIC) WHO Clinical Characterisation Protocol UK (CCP-UK) study of 48,902 hospitalised patients with COVID-19, demonstrating that 85·2% of patients were prescribed antimicrobials at some point of their hospital admission despite positive cultures recorded in less than 3% (*n* = 1107) of patients with only 762 reported secondary infections [9].

Much of the available literature regarding the prevalence of nosocomial co-infections in COVID-19 patients includes all hospital admissions and comes from the early phases of the pandemic [31,32]. Despite the studies containing only critically ill patients reporting a higher prevalence than all hospital admissions [33,34], a systematic review and meta-analysis of 30 studies and 3834 COVID-19 ICU patients still reported that only 14% (95% CI 5–26%) of patients had a co-infection [33]. This further highlights that irrespective of an increased risk of nosocomial infection in all ICU patients [35], for COVID-19 ICU patients, there was a relatively low prevalence of bacterial infection and hence, the requirement for increased vigilance in empirical antimicrobial prescribing [10].

When bacterial secondary infections occurred in COVID-19 patients, they mostly manifested in VAP and new onset bacteraemia in the ICU [36,37]. It has been shown that ICU-acquired bacteraemia rates can be influenced by the inclusion or exclusion of certain organisms. In a retrospective, single-centre study of 78 critically ill patients with COVID-19 reported coagulase-negative staphylococci (11/45) as the most common pathogen in ICU-acquired blood stream infections [38]. We omitted coagulase-negative staphylococci from the positive blood culture inclusion criteria as these, among others, have been shown to demonstrate blood culture contamination, as opposed to true bacteraemia [39]. We found that *Escherichia coli* and *Staphylococcus aureus* were the most common sources of bacteraemia, and *Staphylococcus aureus* was the most common pathogen isolated to sputum cultures, consistent with the ISARIC CCP-UK data [9].

In our study, those with an infection had higher mortality (47.1% in those with an infection vs. 33% in those without an infection). In a similar patient population to ours, recruited from Spain, Andorra, and Ireland, the development of VAP was associated with a significant increase in mortality [40]. In another smaller UK study, ICU-acquired infection and mortality were closely linked [41]. The same investigators also found a positive correlation between the presence of infection and a longer ICU stay (*p* <  0.001). This correlates with an interesting aspect of our findings. 

The length of ICU stay was longer in those who had an infection (median: 12.3 days [IQR 7.9–21]) compared to those without infection (5 days [2.9–7.4]). These results are supported by previous investigations evaluating the outcomes of hospitalised COVID-19 patients with co-infections [42,43,44].

Interestingly, the median length of ITU stay in the No Infection Group correlates with the average first day of infection in the Infected Group (Day 6 [SD 4]). This may indicate that the longer ITU stay in the infected cohort may have been due to the development of their secondary infection. This finding, however, will need to be further corroborated by other data [42,43,44].

An alternative explanation is that patients in the infected group had higher acuity based on their SOFA scores and this led to a prolonged ICU stay. Others have previously described high SOFA scores as a risk factor of secondary infections both in COVID-19 and non-COVID-19 ICU patients [45,46]. The higher burden of organ dysfunction might be associated with different immunomodulatory profiles in COVID-19 and may predispose patients to secondary infectious complications [47]. While the results of our analysis indicate correlations between patient infection status, higher SOFA score, and longer length of ICU stay, we cannot confidently elucidate the causal interplay between them.

There are several potential confounders in the evidence-based contemporary care of COVID-19 ICU patients, which might influence the rate of infection and hence our results. In a post hoc analysis of the UNITE-COVID data set (*n* = 4994), Conway-Morris et al. demonstrated that corticosteroid treatment was strongly associated with the development of ICU-acquired infection following adjustment for identified confounding factors (71% vs. 52%) [36]. Similar results were obtained from other developed countries [48]. All of our patients have received corticosteroids, in line with the recommendations of the RECOVERY trial results, which might have increased their susceptibility to secondary infections [36,49]. 

Furthermore, a prospective study found that treatment with dexamethasone and tocilizumab significantly decreased PCT and CRP values, thereby limiting their ability to track the presence of secondary infections in COVID-19 ICU patients [30].

We have previously also demonstrated the difference in the inflammatory response [50], where the main change in treatment appeared to be the more prevalent use of immunosuppressant medications. The effect of these concomitant pharmacotherapeutic interventions need to be considered when interpreting the utility of PCT values in the ICU patient.

Potential limitations of our study include the retrospective collection of data and the absence of biomarkers on a significant number of days. We have included all consecutive patients to our analysis, reducing the risk of selection bias and used appropriate imputation methods during the analysis. Due to our small sample size and the lack of signal using conventional statistical analysis, we opted against using a multivariate regression model to try to infer associations between PCT levels and infection. We feel that this would have led to an overfitting model with questionable generalisability. Much of the available evidence is limited by the heterogeneity of concomitant treatments in patients over the various waves. In our ICU we had a very high compliance with evidence-based pharmacological treatments, and the vast majority of our patients benefitted from participating in the RECOVERY and REMAP-CAP studies [51,52,53]. Our antimicrobial prescribing was not protocolised, but was guided by at least twice weekly consultant microbiologist ward rounds. All antimicrobial changes were at least discussed on the phone with our colleagues, as is normal practice on our unit. Therefore, any reduction in antimicrobial use is difficult to be attributed to biomarker results.

Overall, despite the initial promise, our results failed to demonstrate PCT as an effective biomarker in tracking the emergence or presence of secondary bacterial infection in COVID-19 ICU patients.

Given the need for effective antimicrobial stewardship in the ICU and the conflicting results regarding traditional protein biomarkers’ role in antimicrobial prescribing in COVID-19 patients, future avenues of research could focus on the efficacy of emerging techniques, such as transcriptomic profiling of the host response and direct identification of pathogens [54,55], which will undoubtedly have increasing roles in the future of antimicrobial treatment strategies.

## 4. Materials and Methods

Our study was a single-centre retrospective observational study at the Grange University Hospital, located within the Aneurin Bevan University Health Board in Wales, UK.

We included patients who were admitted to the ICU between 17 November 2020 to 15 March 2021 (Wave 2), and 3 March 2021 to 22 February 2022 (Wave 3). Other inclusion criteria included being over eighteen years old and a positive nasal pharyngeal PCR test confirming a SARS-CoV-2 infection.

Data were retrospectively collected using patients’ medical notes, accessed through the Aneurin Bevin University Health Board’s Clinical Database. Patient demographics, comorbidities, clinical outcomes, and daily laboratory results were collected up to 14 days after admission to the ICU. WBC count and CRP values were collected daily as part of the standard care in the ICU, with Procalcitonin testing implemented as standard care from 18 March 2020.

Pathogens were identified using blood and sputum cultures collected for microbiological analysis in clinically suspected patients. In general, clinical suspicion of infection was raised if the patients had unexplained high temperature (above 38.3 Celsius), new or increased respiratory secretions, or worsening chest imaging, amongst other clinical signs such as increased need for ventilatory, cardiovascular, or renal support. Patients were classified infected if they had positive microbiological samples from otherwise sterile compartments. 

Sputum samples consisted of the following test types: Bronchoalveolar lavage, non-directed bronchial lavage, and endotracheal secretions. Blood cultures containing common skin contaminants (coagulase-negative staphylococci) were discarded from our results and did not constitute a laboratory-confirmed bloodstream infection (LCBI). Furthermore, cultures with solely fungal pathogens (Candida) were also discarded from our analysis. All patients were tested for pneumococcus antigen and Legionella antigen within 48 h of ICU admission.

### Statistical Analysis

We demonstrated our continuous variables as medians (interquartile ranges) due to the small sample size in our subgroups.

Differences in continuous variables were analysed using independent Mann–Whitney U tests and one-way Analysis of Variances (ANOVA). This allowed us to compare the daily mean PCT values between the two groups: those with a confirmed infection and those without. 

For Delta PCT values, Procalcitonin levels from the day before the first positive culture (t_−1_) and on the day of the first positive culture (t_0_) in the infection cohort, or the day before the first negative culture (t_−1_) and on the day of the first negative culture (t_0_) in the No Infection Group were recorded. The delta PCT was calculated as the changes in the absolute PCT values (subtracting t_−1_ from t_0_); the percentages were calculated as [(t_0_ − t_−1_)/t_−1 *×*_ 100]. Missing values were imputed using the multiple imputation method with predictive mean matching.

The antibiotic density calculated in this study represents the number of days the patients were on each antibiotic during their stay in the ICU. This means if patients were on two antibiotics for two days, they would have a cumulative four days on antibiotics. Their total number of days in the ICU was divided by the cumulative number of days on antibiotics.

Data were collected using Microsoft Excel Version 16.62. Statistical analysis was performed using IBM SPSS Statistics 27 and RStudio Version 2022.12.0+353. Data visualisation and graph formation were undertaken with GraphPad Prism Version 9.5.1. A *p*-value < 0.05 indicated statistical significance.

## 5. Conclusions

We demonstrated that both absolute and delta PCT values failed to indicate the presence of secondary bacterial infection in COVID-19 ICU patients. Our data support current NICE guidance [25] in avoiding routine procalcitonin testing for the guidance of antimicrobial therapy in hospitalised COVID-19 patients.

## Figures and Tables

**Figure 1 antibiotics-12-00709-f001:**
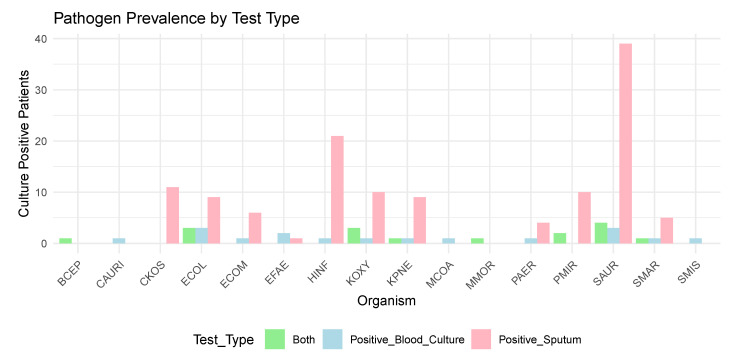
Prevalence of Pathogens Detected by Blood Cultures and Sputum Cultures (see Appendix A for Microbiology KEY).

**Figure 2 antibiotics-12-00709-f002:**
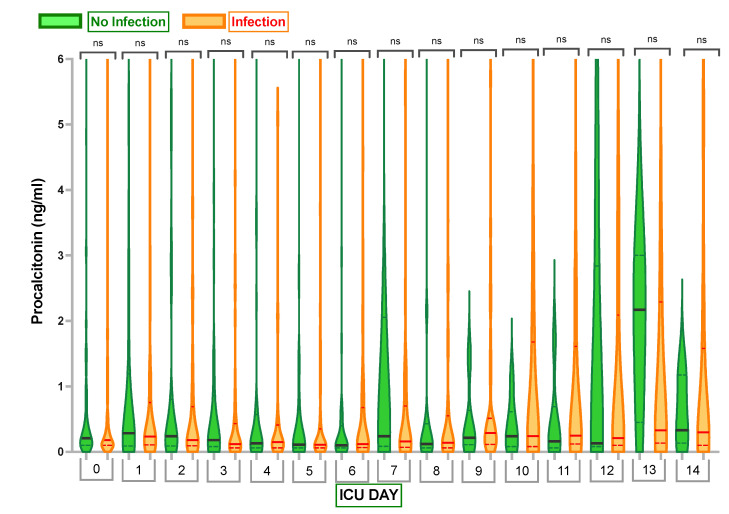
Comparison of Procalcitonin (PCT) Levels in Infection Group and No Infection Group. Violin plots represent the distribution of data. Hard lines show medians, and dashed lines show interquartile ranges. The *y*-axis represents Procalcitonin Levels in ng/mL. The *x*-axis represents Days in the Intensive Care Unit (ICU) with 0—Procalcitonin Level Day 0. ns—no statistically significant difference from the Independent Mann–Whitney U test or one-way analysis of variances (ANOVA).

**Figure 3 antibiotics-12-00709-f003:**
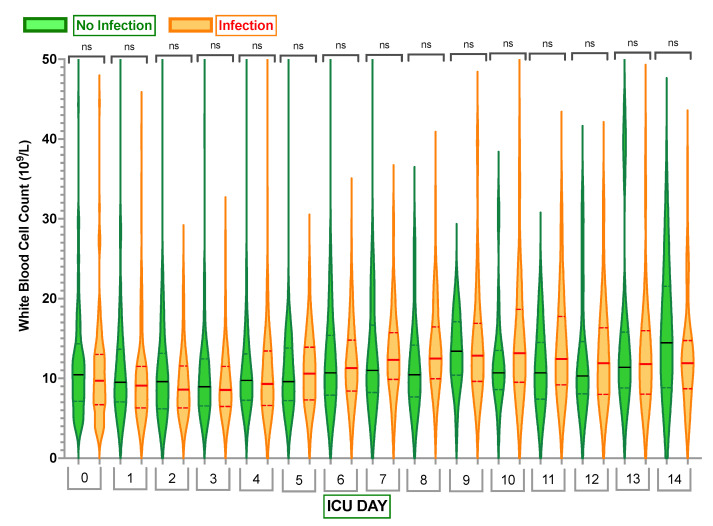
Comparison of white blood cell (WBC) count in the Infection Group and No Infection Group. Violin plots represent the distribution of data. Hard lines show medians, and dashed lines show interquartile ranges. The y-axis represents white blood cell count in 10^9^/L. The x-axis represents Days in the Intensive Care Unit (ICU) with 0—white blood cell count Day 0. ns—no statistically significant difference from the independent Mann–Whitney U test.

**Figure 4 antibiotics-12-00709-f004:**
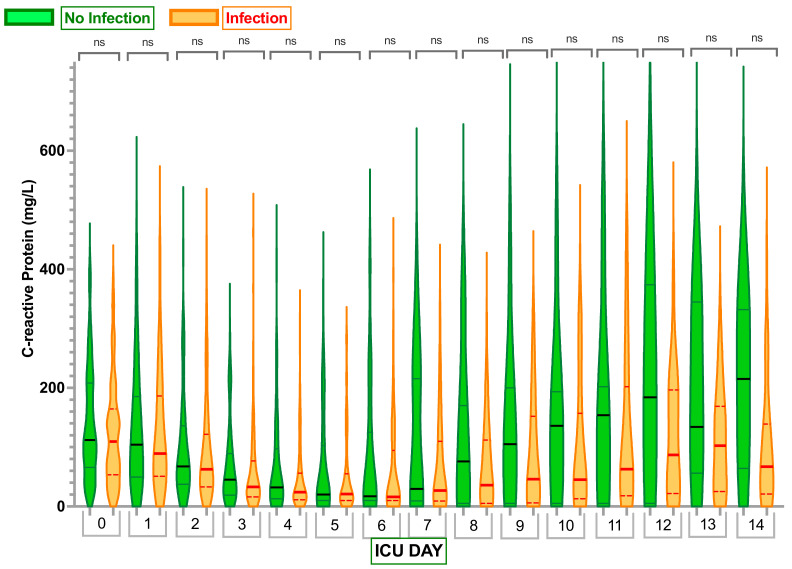
Comparison of C-reactive protein (CRP) levels in Infection Group and No Infection Group. Violin plots represent the distribution of data. Hard lines show medians, and dashed lines show interquartile ranges. The y-axis represents C-reactive protein (CRP) levels in mg/L. The x-axis represents Days in the Intensive Care Unit (ICU) with 0—C-reactive protein level Day 0. ns—no statistically significant difference from the Independent Mann–Whitney U test.

**Figure 5 antibiotics-12-00709-f005:**
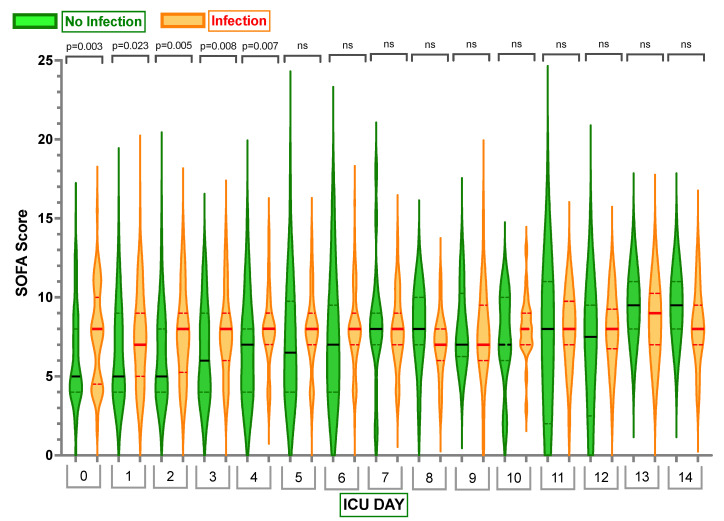
Comparison of Sequential Organ Failure Assessment (SOFA) scores in Infection Group and No Infection Group. Violin plots represent the distribution of data. Hard lines show medians, and dashed lines show interquartile ranges. The y-axis represents SOFA Scores in Days. The x-axis represents Days in the Intensive Care Unit (ICU) with 0—SOFA Score Day 0. ns—no statistically significant difference from the Independent Mann–Whitney U test. *p* < 0.05 indicates statistical significance.

**Figure 6 antibiotics-12-00709-f006:**
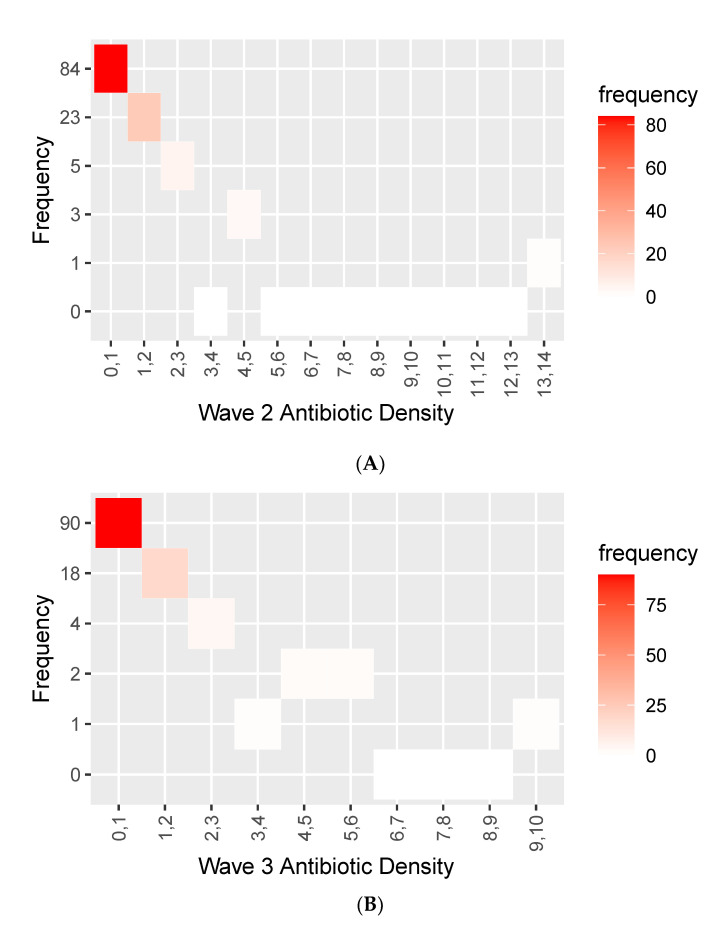
Histogram showing antibiotic/antimicrobial density during (**A**) Wave 2 and (**B**) Wave 3.

**Figure 7 antibiotics-12-00709-f007:**
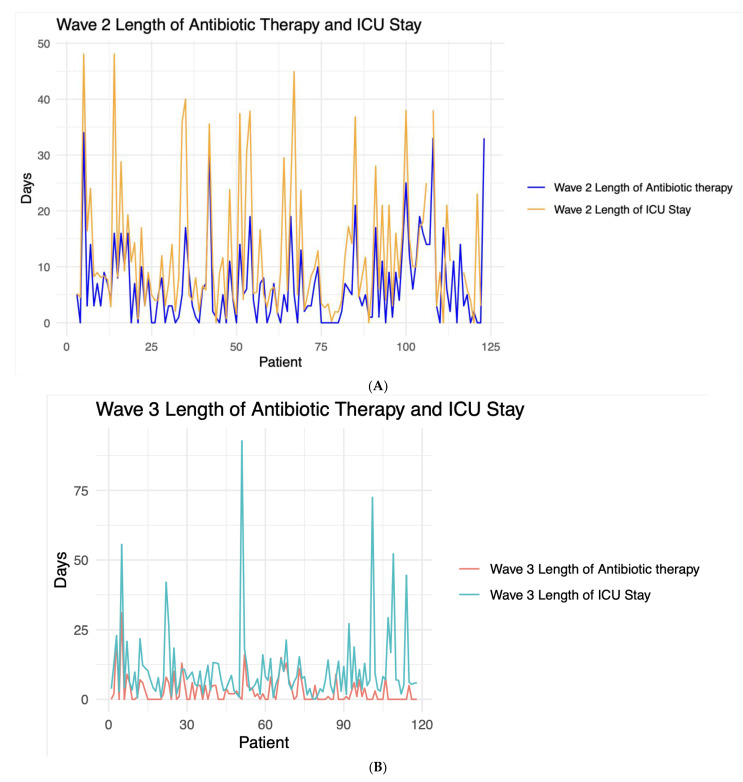
Line graph contrasting the length of antibiotic/antimicrobial therapy and length of Intensive Care Unit (ICU) stay for each patient during (**A**) Wave 2 and (**B**) Wave 3.

**Figure 8 antibiotics-12-00709-f008:**
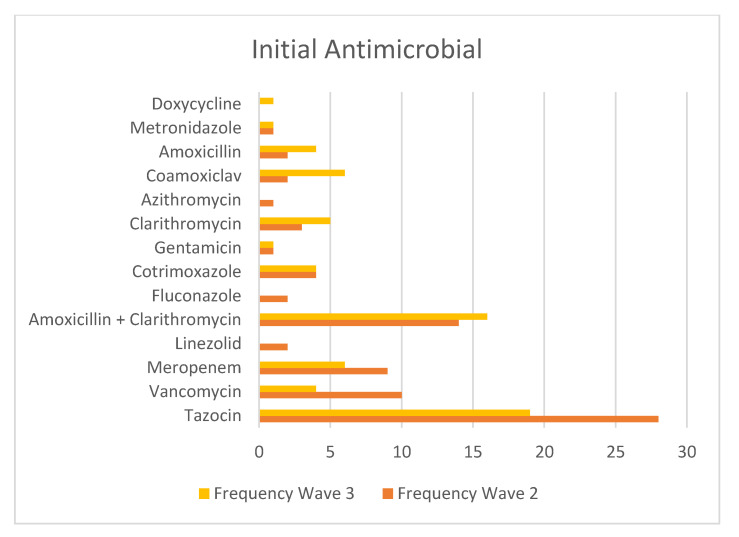
Table showing the frequency of each initial antimicrobial from both waves.

**Table 1 antibiotics-12-00709-t001:** Comparison of Patient Demographics, Comorbidities, SOFA Scores, Length of Stay in ICU, and Mortality between those who had an infection and those who did not.

	Count (*n*) (No Infection/ Infection)	No Infection (*n* = 115)	Infection (*n* = 119)	Overall (*n* = 234)
**Age (years)**				
Mean (SD)	(115/119)	56.58 (14.33)	55.08 (12.386)	55.81 (13.37)
Median [Min, Max]		60 [28, 80]	57 [23, 76]	58 [23, 80]
**Sex (Male/Female, *n***)	(115/119)	87/28	85/34	172/62
**Outcome**	(115/119)			
Alive (*n*, %)		76 (66.7%)	63 (52.9%)	139 (59.7%)
Deceased (*n*, %)		38 (33%)	56 (47.1%)	94 (40.3%)
**Comorbidities**	(115/119)			
Diabetes		28 (24.6%)	29 (24.4%)	57 (24.4%)
Hypertension		33 (28.9%)	42 (35.3%)	75 (32.2%)
Cardiovascular disease		16 (14.2%)	16 (13.4%)	233 (13.8%)
COPD		8 (7%)	3 (2.5%)	11 (4.7%)
Asthma		18 (15.8%)	26 (21.8%)	44 (18.9%)
Renal Impairment		11 (9.6%)	4 (3.4%)	15 (6.4%)
**SOFA score Day 0**	(91/97)			
Mean (SD)		6.22 (3.1)	7.62 (3.06)	6.95 (3.14)
Median [Min, Max]		5 [0, 15]	8 [2, 16]	7 [0, 16]
**Length of ICU stay (days)**	(115/119)			
Mean (SD)		6.15 (6.2)	16.38 (13.1)	11.29 (11.47)
Median [Min, Max]		5 [0.08, 52.2]	12.3 [0, 72.5]	7.5 [0, 72.5}

**Table 2 antibiotics-12-00709-t002:** Median, Q1 (0.25), and Q3 (0.75) Procalcitonin (PCT) Values from Day 0 (PCT 0) to Day 14 (PCT 14) in those who had an infection and those who did not.

	Infection	0.25	Median	0.75	Count (*n*)
Day of admission ng/mL	no infection	0.10	0.23	0.77	115
	Infection	0.10	0.18	0.54	119
Day 1 ng/mL	no infection	0.09	0.30	1.26	112
	Infection	0.11	0.23	0.75	112
Day 2 ng/mL	no infection	0.10	0.24	0.81	105
	Infection	0.09	0.18	0.69	111
Day 3 ng/mL	no infection	0.08	0.18	0.86	91
	Infection	0.06	0.12	0.43	110
Day 4 ng/mL	no infection	0.06	0.13	0.57	76
	Infection	0.06	0.15	0.41	106
Day 5 ng/mL	no infection	0.06	0.11	0.64	65
	Infection	0.06	0.11	0.35	100
Day 6 ng/mL	no infection	0.06	0.10	0.40	49
	Infection	0.07	0.12	0.68	96
Day 7 ng/mL	no infection	0.09	0.24	2.05	37
	Infection	0.07	0.16	0.70	94
Day 8 ng/mL	no infection	0.06	0.12	0.43	27
	Infection	0.06	0.14	0.55	88
Day 9 ng/mL	no infection	0.11	0.22	0.63	19
	Infection	0.12	0.29	0.51	79
Day 10 ng/mL	no infection	0.08	0.24	0.61	16
	Infection	0.08	0.24	1.68	72
Day 11 ng/mL	no infection	0.06	0.16	0.69	12
	Infection	0.12	0.25	1.61	68
Day 12 ng/mL	no infection	0.08	0.13	2.84	11
	Infection	0.10	0.21	2.09	63
Day 13 ng/mL	no infection	0.45	2.17	2.91	9
	Infection	0.14	0.33	2.29	57
Day 14 ng/mL	no infection	0.14	0.33	1.17	5
	Infection	0.10	0.30	1.58	54

## Data Availability

The data presented in this study are available on request from the corresponding author.

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
