# Peer review of "Procalcitonin Values Fail to Track the Presence of Secondary Bacterial Infections in COVID-19 ICU Patients"

_antibiotics, 2023, doi:10.3390/antibiotics12040709_

Round 1

Reviewer 1 Report

Elsa Harte and coworkers conducted a retrospective analysis on 238 patients in order to assess the value of procalcitonin in identifying secondary infections in Covid-19 patients (during the second and third wave) and, consequently, guiding antimicrobial prescribing in ITU patients.

This subject has a real impact in clinical practice because there is a great need for reliable biomarkers that are capable to help initiate and stop antibiotics.

This study come with results similar with some previous reports and show  no statistically significant difference between inflammatory biomarkers levels (PCT, WBC or CRP) in those with an associated infection versus those without infection.

I agree to the publication of the material.

Author Response

Thank you very much for the kind words. We are happy that the Reviewer found our data worthwhile and publishable.

Reviewer 2 Report

The authors demonstrate lack of utility of procalcitonin in predicting secondary bacterial infections in COVID-19 ICU patients. Study is well written and correctly notes that use of immunomodulatory therapy in COVID-19 patients could be playing a role. Please note some suggestions below

- Did the authors do separate analysis of waves 2 and 3. As antibiotic use was different during each wave; it would be interesting to see if predictability of procalcitonin will change.

- in lines 291-298, the statements on prevalence of bacterial infections in ICU patients is contradictory. Initially it is mentioned there is high prevalence of infections in ICU patients, but the next paragraph says low prevalence. This needs to be clarified. 

Author Response

The authors demonstrate lack of utility of procalcitonin in predicting secondary bacterial infections in COVID-19 ICU patients. Study is well written and correctly notes that use of immunomodulatory therapy in COVID-19 patients could be playing a role. Please note some suggestions below

- Did the authors do separate analysis of waves 2 and 3. As antibiotic use was different during each wave; it would be interesting to see if predictability of procalcitonin will change.

A: Thank you for your suggestion. Yes, we also did a separate analysis of waves 2 and 3 prior to combining the 2 waves- the results in both waves were similar. We do not feel it would add more to the study, however if the Editor would prefer we would be happy to supply this data as a Supplementary Appendix.

- in lines 291-298, the statements on prevalence of bacterial infections in ICU patients is contradictory. Initially it is mentioned there is high prevalence of infections in ICU patients, but the next paragraph says low prevalence. This needs to be clarified. 

A: Thank you for the observation. We tried to clarified this apparent contradiction in the revised manuscript:

Much of the available literature regarding the prevalence of nosocomial co-infections in COVID-19 patients includes all hospital admissions and comes from the early phases of the pandemic [31, 32]. Despite the studies containing only critically ill patients reporting a higher prevalence than all hospital admissions [33, 34], a systematic review and meta-analysis of thirty studies and 3834 COVID-19 ICU patients still reported that only 14% (95% CI 5-26%) of patients had a co-infection [33]. This further highlights that irrespective of an increased risk of nosocomial infection in all ICU patients [35], for COVID-19 ICU patients, there was a relatively low prevalence of bacterial infection and hence, the requirement for increased vigilance in empirical antimicrobial prescribing [10].

Reviewer 3 Report

I have reviewed with interest the article "Can Procalcitonin Values Track the Presence of Secondary Infections in COVID-19 ICU Patients?" submitted to "Antibiotics" and the forthcoming special issue "Antimicrobial Resistance during the COVID-19 Outbreak". The study hospital, a new 500-bed Grange University hospital in Wales, opened in 2020 one year ahead of schedule to alleviate the care of Covid-19 patients. The authors have chosen to retrospectively investigate ICU patients with documented covid -19 infections from the pandemic's second and third waves (2021-22). They have evaluated the utility of daily measured procalcitonin, CRP and WBC for two cohorts, with and without confirmed non-viral co-infection, and measured the antibiotic use.

Title:  Using a question as a title is generally not considered good practice.
I suggest changing it to reflect what you investigated and the main conclusion. That is, failing to demonstrate that procalcitonin is suited as a marker for bacterial co-infections.

 Main comments and questions:

 1)      Inclusion of the patients & their status on admittance: you included adult patients ( > 18 y) admitted at your ICU, but there is no mention or tabulation ("Infection / No infection") of the history of these patients before the admission:

  a) In what proportions were they admitted from their homes, nursing homes, other hospitals, and
  b) Before inclusion, how long had they been treated in any external institutions or your own (non-ICU) departments?

 2)      Use of the SOFA score determines the extent of a patient's organ function or rate of failure and is mainly a sepsis assessment score predicting likely outcomes in septic patients. (You are, of course, referring to the full SOFA score and not the qSOFA replacement for SIRS criteria). While Covid-19 certainly is a systemic disease, respiratory failure is the principal cause for ICU admittance in these cases, I presume all included patients needed mechanical ventilation (please state). 

3)      The list of comorbidities in Table 1 shows some conditions that seem somewhat irrelevant to the topic under investigation, namely high cholesterol and anticoagulants (which is a medication, not a comorbidity). What about immunocompromised patients (not excluded from this study, although discouraged from being targets for PCT testing according to NICE recommendations). These are also highly susceptible to co-infections. Please comment.

4)      You state in the Discussion (lines 237-9)"…the similarity in PCT levels between cohorts support the current 237 NICE COVID-19 Guidance NG191". The NICE guidelines do not advise against identifying a bacterial infection but say: "There is insufficient evidence to recommend routine procalcitonin testing to guide decisions about antibiotics. Centres already using procalcitonin tests are encouraged to participate in research and data collection. Procalcitonin tests could be useful in identifying whether there is a bacterial infection. However, it is not clear whether they add benefit beyond what is suggested in the recommendation on tests to help differentiate between viral and bacterial pneumonia to guide decisions about antibiotics." To this end, NICE recommend the following (citation):

"

·         a full blood count

·         chest imaging (X-ray, CT or ultrasound)

·         respiratory and blood samples (for example, sputum or a tracheal aspirate sample, blood culture)

·         urine samples for legionella and pneumococcal antigen testing

·         throat, nasopharyngeal or sputum samples for respiratory viral (and atypical pathogen) polymerase chain reaction testing

"

5)      I am concerned with the sparse information in the Methods section and Appendix A on the total diagnostic approach to and reporting bacterial co-infections. The negative conclusions of your study will rely heavily upon your stated criteria for determining a bacterial RTI / sepsis from retrospective data. An example is your statement in lines 253-55; that your finding of low median absolute PCT values renders the test unfit to determine bacterial etiology.

Referring to the NICE recommendations cited in comment 4), my questions are:      
  a) you do not give any results of chest imaging as part of the infections evaluation of your patients. Were there differences in the use of imaging procedures? Did chest CT results differ, as covid-19 infiltrates might be distinguished fom those caused by S. aureus? 
  b)  You write that "sputum samples" were cultivated; were these tube secretes, samples collected by the use of BAL or otherwise from deeper airway regions?
  c) Were legionella and pneumococcal antigens tested? Atypical pathogen testing

6)      I have several comments and questions on the bacteriology reporting:

  a) Only candida and coagulase-negative staphylococci were excluded and should therefore be removed from the pathogens list in the Appendix A (and mixed CoN-S, "MCOA" also from Figure 1). Aspergillosis, treated with azoles and not antibacterial compounds, should perhaps also be considered removed?
Furthermore, your list of pathogens should display the number of culture-positive patients per agent for better transparency.  
  b) Is the finding in the sputum samples of every microorganism listed and the fact that a sample was obtained on suspicion of infection, together considered proof of an infection? This must be clarified, not least since CRP and WBC did not differ significantly between the infected and non-infected groups.
  c) You are discussing the longer stay in the ICU for the infection versus the non-infection group and speculate that this "may have been due to the development of their secondary infection", proposing that this fact would have to be "further corroborated in studies" (lines 324-7). Perhaps an alternative interpretation could be that a more extended stay in the ICU may be a result of more heavily colonized (and infected) patients, while the cause is more severe underlying diseases in these patients - of which we know little from your list (Table 1) of these conditions (see my comment 3). Longer ICU stays in the infection group may further be explained by the higher SOFA scores, indicating a more severe acute (covid-19) disease (see my comment 2).
Please include some consideration to this in your article after supplementing the requested data.
  d) In all, 39 respiratory samples (all "sputum") grew S. aureus, but only three also with a positive blood culture. In VAP/nosocomial infections, SA may well be a significant pathogen but not so in community-acquired infections. Colonizing SA are often considered clinically non-significant also in a ICU setting, partly dependent on the sampling quality (see also my comment 5a). As per antibiogram results, one would expect the use of cloxacillin but only vancomycin and linezolid (as well as piperacillin/tazobactam/meropenem) is mentioned. Is this because of a very high prevalence of MRSA in your institution?

7)      Some concerns regarding the measurement for antibiotic use:

  a) Please define the antibiotic use measurement in the Methods section, as "antibiotic density" is now explained only in the text in the Table 6 and is not so easy to find for the casual reader.
  b) In surveillance, your "antibiotic density" is denoted with the
DOT metric: "Days of therapy" is the sum of calendar days in which at least a single dose of antibiotic is received, and in combination regimes, each antibiotic counts as a day.
However, LOT or" Length of therapy" (
the number of calendar days of antibiotic treatment regardless of the number of agents used) may be regarded a more appropriate measure since a reduction in treatment duration is the purpose of PCT as a decision aid in antibiotic stewardship.  

Language, layout, misc:

1)      You use two abbreviations for Intensive Care Unit throughout the manuscript: ITU and ICU. Please use only the latter.

2)      Line 48-- : What does "This " refer to? Please rephrase since "this" points to the preceding sentence - i.e. the disparity revealed in studies does not drive antibiotic resistance; the failure to recognize a bacterial etiology in a co-infection is. 

3)      Table 2 needs a column which gives the number of patients for each row, Infection / No infection, since the median values for LOS in the ICU are 16,4 and 4,2, respectively.

4)      It may be only me, but the "violin plot" in Figures 2-4 is unfamiliar. A quick google search helped. But I still do not understand what the many colour nuances in the "violins" represent, as the labelling only shows two distinct colours. This is confusing; please explain.

5)      Lines 373-4  have already been stated in the "Informed Consent Statement" at the end of the article and should be omitted here.

Author Response

Title:  Using a question as a title is generally not considered good practice. 
I suggest changing it to reflect what you investigated and the main conclusion. That is, failing to demonstrate that procalcitonin is suited as a marker for bacterial co-infections.

A: thanks you for this comment. We took on board the suggestion and changed the title as follows:

Procalcitonin Values Fail to Track the Presence of  Secondary Bacterial Infections in COVID-19 ICU Patients

 Main comments and questions:

 1)      Inclusion of the patients & their status on admittance: you included adult patients ( > 18 y) admitted at your ICU, but there is no mention or tabulation ("Infection / No infection") of the history of these patients before the admission:

  1. In what proportions were they admitted from their homes, nursing homes, other hospitals, and 
    b) Before inclusion, how long had they been treated in any external institutions or your own (non-ICU) departments?

A: Thank you for these important questions. All patients were admitted from their homes, 13 patients (5.9%) were transferred from other ICUs as capacity transfer. Median length of hospital stay was 3 (1-5) days before ICU admission. We have added this information to the Results.

 2)      Use of the SOFA score determines the extent of a patient's organ function or rate of failure and is mainly a sepsis assessment score predicting likely outcomes in septic patients. (You are, of course, referring to the full SOFA score and not the qSOFA replacement for SIRS criteria). While Covid-19 certainly is a systemic disease, respiratory failure is the principal cause for ICU admittance in these cases, I presume all included patients needed mechanical ventilation (please state). 

A: Thank you for the comment. We used the full SOFA score to evaluate our patients. The reviewer is right, all patients were mechanically ventilated.

3)      The list of comorbidities in Table 1 shows some conditions that seem somewhat irrelevant to the topic under investigation, namely high cholesterol and anticoagulants (which is a medication, not a comorbidity). What about immunocompromised patients (not excluded from this study, although discouraged from being targets for PCT testing according to NICE recommendations). These are also highly susceptible to co-infections. Please comment.

A: Thanks for this observation. We agree with the reviewer that some of the listed variables are superfluous. We have omitted high cholesterol and patients who were treated with anticoagulants. In our cohort we did not have any patients with severe underlying immunocompromise.

4)      You state in the Discussion (lines 237-9)"…the similarity in PCT levels between cohorts support the current 237 NICE COVID-19 Guidance NG191". The NICE guidelines do not advise against identifying a bacterial infection but say: "There is insufficient evidence to recommend routine procalcitonin testing to guide decisions about antibiotics. Centres already using procalcitonin tests are encouraged to participate in research and data collection. Procalcitonin tests could be useful in identifying whether there is a bacterial infection. However, it is not clear whether they add benefit beyond what is suggested in the recommendation on tests to help differentiate between viral and bacterial pneumonia to guide decisions about antibiotics." To this end, NICE recommend the following (citation):

"

  • a full blood count
  • chest imaging (X-ray, CT or ultrasound)
  • respiratory and blood samples (for example, sputum or a tracheal aspirate sample, blood culture)
  • urine samples for legionella and pneumococcal antigen testing
  • throat, nasopharyngeal or sputum samples for respiratory viral (and atypical pathogen) polymerase chain reaction testing

"

A: Thanks for the comment. We don’t think our data would suggest otherwise, however to make clearer point we have amended the Discussion:

The current NICE COVID-19 Guidance NG191 emphasises the current insufficiency of evidence to recommend routine PCT testing as a method of guiding antimicrobial prescribing [25]. Our results demonstrating the similarity in PCT levels between infected and non-infected cohorts further adds to the evidence and suggest its use does not add benefit in identifying bacterial infections and subsequent decisions regarding antimicrobial prescribing.

5)      I am concerned with the sparse information in the Methods section and Appendix A on the total diagnostic approach to and reporting bacterial co-infections. The negative conclusions of your study will rely heavily upon your stated criteria for determining a bacterial RTI / sepsis from retrospective data. An example is your statement in lines 253-55; that your finding of low median absolute PCT values renders the test unfit to determine bacterial etiology.

Referring to the NICE recommendations cited in comment 4), my questions are:      
  a) you do not give any results of chest imaging as part of the infections evaluation of your patients. Were there differences in the use of imaging procedures? Did chest CT results differ, as covid-19 infiltrates might be distinguished fom those caused by S. aureus?  

A: Thanks for these comments. Diagnosis of Ventilator Associated Pneumonia is fraught with difficulties and inadequacies. The NICE Guidance, although mentines the potential diagnostic steps leaves this to the clinicians. To note, we have previously shown that the lack of chest imaging can indeed influence the reported rates of VAP (Pugh et al 2016 Frontiers in Microbiology https://www.frontiersin.org/articles/10.3389/fmicb.2016.01271/full). Most of our patients in the study had a CT scan on admission to the ICU and some of them had repeated scans during their course. We have not collected this data as it would have been too cumbersome and difficult to interpret in the context of COVID-19. Our data implies, that on some days PCT levels were higher in the non-infected group, that is where we could not confirm bacterial infection using microbiology techniques. We acknowledge, that culture-negative VAP is common in ventilated patients, however our data shows that PCT was not a useful marker in this context either.

  1. b)  You write that "sputum samples" were cultivated; were these tube secretes, samples collected by the use of BAL or otherwise from deeper airway regions? 

A: Thanks you for pointing out this omission. The sputum samples were indeed collected using either directed BAL or more commonly NBAL techniques or endotracheal aspiration.

  1. c) Were legionella and pneumococcal antigens tested? Atypical pathogen testing

A: All patients had pneumococcus and legionella antigen testing performed within 24 hours of ICU admission and the results demonstrated no incidence of either pathogen.

6)      I have several comments and questions on the bacteriology reporting:

  1. a) Only candida and coagulase-negative staphylococci were excluded and should therefore be removed from the pathogens list in the Appendix A (and mixed CoN-S, "MCOA" also from Figure 1). Aspergillosis, treated with azoles and not antibacterial compounds, should perhaps also be considered removed? 
    Furthermore, your list of pathogens should display the number of culture-positive patients per agent for better transparency.  

A: Thank you for this important point, we agree with the reviewer that these could be removed and we have changed the list in Appendix A and updated the graph as requested.

  1. Is the finding in the sputum samples of every microorganism listed and the fact that a sample was obtained on suspicion of infection, together considered proof of an infection? This must be clarified, not least since CRP and WBC did not differ significantly between the infected and non-infected groups.

A: Thank you for these questions. Yes, we have listed every organism found in respiratory samples (no selection bias). Our current clinical practice is that we initiate microbiological sampling on clinical suspicion of infection. If the sample becomes positive we consider it as a proof of an infection as per the ECDC criteria for healthcare associated infections.

  1. c) You are discussing the longer stay in the ICU for the infection versus the non-infection group and speculate that this "may have been due to the development of their secondary infection", proposing that this fact would have to be "further corroborated in studies" (lines 324-7). Perhaps an alternative interpretation could be that a more extended stay in the ICU may be a result of more heavily colonized (and infected) patients, while the cause is more severe underlying diseases in these patients - of which we know little from your list (Table 1) of these conditions (see my comment 3). Longer ICU stays in the infection group may further be explained by the higher SOFA scores, indicating a more severe acute (covid-19) disease (see my comment 2). 
    Please include some consideration to this in your article after supplementing the requested data.

A: Thanks for these comments. We have supplied the requested data and we feel even more confident that our original concept was possibly correct. We agree that higher acuity may lead to longer ICU stay and we have reflected on this in the updated Discussion.

Interestingly, the median length of ITU stay in the No Infection Group correlates with the average 1st day of infection in the Infected Group (Day 6 [SD 4]). This may indicate that the longer ITU stay in the infected cohort may have been due to the development of their secondary infection. This finding, however, will need to be further corroborated by other data [42-44].

An alternative explanation is that patients in the infected group had higher acuity based on their SOFA scores and this led to prolonged ICU stay. Others have previously described high SOFA scores as a risk factor of secondary infections both in COVID-19 and non-COVID-19 ICU patients [45,46]. Higher burden of organ dysfunction might be associated with different immunmodulatory profiles in COVID-19 and may predispose patients to secondary infectious complications [47]. While the results of our analysis indicate correlations between patient infection status, higher SOFA score, and longer length of ICU stay, we cannot confidently elucidate the causal interplay between them.

  1. d) In all, 39 respiratory samples (all "sputum") grew S. aureus, but only three also with a positive blood culture. In VAP/nosocomial infections, SA may well be a significant pathogen but not so in community-acquired infections. Colonizing SA are often considered clinically non-significant also in a ICU setting, partly dependent on the sampling quality (see also my comment 5a). As per antibiogram results, one would expect the use of cloxacillin but only vancomycin and linezolid (as well as piperacillin/tazobactam/meropenem) is mentioned. Is this because of a very high prevalence of MRSA in your institution?

A: Thank you for this observations. We agree that S. aureus could be either a pathogen or a contaminant. It’s almost impossible to distinguish between the two, however it has been noted that this was the most common pathogen in similar UK settings. The reviewer is right to question the use of antimicrobials and this has been a topic of ongoing discussion during the second and third wave of the pandemic. Ultimately, antimicrobial regimes were based on advice from clinical microbiologists, who visited the ICU three times every week and discussed all positive culture results with the ICU consultants. Our antimicrobial guidelines suggest the use of vancomycin or linezolid in these situations, despite the very low prevalence (less than 0.5%) of MRSA. Apologies for the lack of clarity regarding the pathogen prevalence graph- the value 39 related to S. aureus refers to the number of patients that were positive for the pathogen: ie 39 patients had positive sputum culture for S. Aureus, 3 patients had positive blood culture for S. Aureus, and 4 patients had both positive blood and sputum culture for S. Aureus.

7)      Some concerns regarding the measurement for antibiotic use:

  1. a) Please define the antibiotic use measurement in the Methods section, as "antibiotic density" is now explained only in the text in the Table 6 and is not so easy to find for the casual reader. –

A: Thanks for this pointer, we agree and it has been corrected.

  1. b) In surveillance, your "antibiotic density" is denoted with the DOT metric: "Days of therapy" is the sum of calendar days in which at least a single dose of antibiotic is received,and in combination regimes, each antibiotic counts as a day.
    However, LOT or" Length of therapy" (the number of calendar days of antibiotic treatment regardless of the number of agents used) may be regarded a more appropriate measure since a reduction in treatment duration is the purpose of PCT as a decision aid in antibiotic stewardship. 

A: Thank you for this very valid point. We have subsequently calculated each patient's Length of Therapy (number of calendar days of antibiotic treatment regardless of the number of agents used) as requested and produced the following graphs for further clarity:

Language, layout, misc:

  • You use two abbreviations for Intensive Care Unit throughout the manuscript: ITU and ICU. Please use only the latter. 

A: Thank you, we corrected this.

2)      Line 48-- : What does "This " refer to? Please rephrase since "this" points to the preceding sentence - i.e. the disparity revealed in studies does not drive antibiotic resistance; the failure to recognize a bacterial etiology in a co-infection is.  

A: Corrected.

However, studies have subsequently revealed a significant disparity between the high rate of antimicrobial therapy and the low prevalence of confirmed secondary bacterial infections [9,10]

The difficulty in recognizing a true co-infection of bacterial aetiology may have resulted in an unnecessary increase in antimicrobial prescribing [11,12], further exacerbating an antimicrobial resistance crisis that was considered a threat to modern medicine even prior to the pandemic [13]. The clinical community proposed further research into the various methods of identifying secondary bacterial infections in COVID-19 patients, thus determining optimal strategies of antimicrobial prescribing.

  • Table 2 needs a column which gives the number of patients for each row, Infection / No infection, since the median values for LOS in the ICU are 16,4 and 4,2, respectively.

A: Thank you the numbers at risk values are now provided.

4)      It may be only me, but the "violin plot" in Figures 2-4 is unfamiliar. A quick google search helped. But I still do not understand what the many colour nuances in the "violins" represent, as the labelling only shows two distinct colours. This is confusing; please explain.

A: The violin plots are considered as a better alternative to standard box-and-whisker plots as they better represent the distribution of the data, whilst maintaining the visualization of important central tendencies. The colour scheme was an auto function of the software for an aesthetic effect, and there is no underlying nuance of the colour change- please find the updated graphs with the colour change removed.

5)      Lines 373-4  have already been stated in the "Informed Consent Statement" at the end of the article and should be omitted here.

A: Thanks for the suggestion, we have removed the sentences.

Round 2

Reviewer 3 Report

I find that the changes in the manuscript and the answers to my questions  satisfactory. A fine contribution to the (extensive) research on the use of procalcitonin in hospital care.